# Direct detection of atomic oxygen on the dayside and nightside of Venus

**Heinz-Wilhelm Hübers** [1,2] ✉, **Heiko Richter** [1,6], **Urs U. Graf** [3], **Rolf Güsten** [4], **Bernd Klein** [4,5], **Jürgen Stutzki** [3] & **Helmut Wiesemeyer** [4]

Atomic oxygen is a key species in the mesosphere and thermosphere of Venus. It peaks in the transition region between the two dominant atmospheric circulation patterns, the retrograde super-rotating zonal flow below 70 km and the subsolar to antisolar flow above 120 km altitude. However, past and current detection methods are indirect and based on measurements of other molecules in combination with photochemical models. Here, we show direct detection of atomic oxygen on the dayside as well as on the nightside of Venus by measuring its ground-state transition at 4.74 THz (63.2 μm). The atomic oxygen is concentrated at altitudes around 100 km with a maximum column density on the dayside where it is generated by photolysis of carbon dioxide and carbon monoxide. This method enables detailed investigations of the Venusian atmosphere in the region between the two atmospheric circulation patterns in support of future space missions to Venus.

Atomic oxygen is produced on the dayside of Venus by photolysis of carbon dioxide ($CO_2$) and carbon monoxide (CO). It is transported to the nightside by the subsolar to antisolar circulation, where it accumulates near the antisolar point[1–3]. On the nightside, atomic oxygen recombines in a three-body reaction to molecular oxygen. Atomic oxygen is an important species for the photochemistry, because it is very abundant in the mesosphere and lower thermosphere and interacts with many other molecules such as oxygen ($O_2$), CO, and $CO_2$[3]. Also, it is important for the energy balance, because $CO_2$ is collisionally excited by atomic oxygen and the $CO_2$ 15-μm emission is the dominant cooling mechanism of the mesosphere and thermosphere[4,5]. In addition, atomic oxygen can be used as tracer for the global circulation in the upper thermosphere (approximately 130–250 km) as demonstrated by measurements of the oxygen dayglow with the Pioneer Venus Orbiter[6]. Despite its importance, direct observations of atomic oxygen in the mesosphere and lower thermosphere are scarce. The only direct detection has been made by observation of its 557-nm green line in the night airglow of Venus[7,8]. This observation is limited to the nightside and little variability of the atomic oxygen concentration from the center of Venus towards the edge was found. Atomic oxygen density maps of the Venus nightside at altitudes between 90 and 120 km have been derived indirectly from observations of the $O_2$ night airglow at 1.27 μm by the Visible and InfraRed Thermal Imaging Spectrometer (VIRTIS) onboard the Venus Express (VEX) satellite[1,2,9]. This method requires global maps of the $O_2$ 1.27-μm emission originating from the $O_2(^1\Delta) \rightarrow O_2(^3\Sigma)$ transition, of the $CO_2$ density and of the temperature. In combination with a photochemical model, which is based on a-priori information such as quenching coefficients, reaction rates and efficiencies, atomic oxygen densities have been calculated[1,2]. These observations show that the atomic oxygen density peaks around 104 km near the antisolar point. A decrease of 30–40% is observed from 00:00 local time (LT) to 22:00 LT and towards ±30° latitude along the midnight meridian[1]. Qualitatively, the maps and vertical profiles derived from VIRTIS data agree with simulations although the decrease away from the antisolar point is somewhat stronger (approximately 30%) in the models than observed by VIRTIS[1,2,10,11].

Here, we present measurements of atomic oxygen on the dayside as well as on the nightside. We have used the upGREAT (German Receiver for Astronomy at Terahertz Frequencies) array spectrometer[12] on the Stratospheric Observatory for Infrared Astronomy (SOFIA) (see methods, subsection "SOFIA/upGREAT observations")[13]. The observations were made in the early evenings of

[1]Deutsches Zentrum für Luft- und Raumfahrt (DLR), Institute of Optical Sensor Systems, Berlin, Germany. [2]Humboldt-Universität zu Berlin, Department of Physics, Berlin, Germany. [3]I. Physikalisches Institut der Universität zu Köln, Köln, Germany. [4]Max-Planck-Institut für Radioastronomie, Bonn, Germany. [5]University of Applied Sciences Bonn-Rhein-Sieg, Sankt Augustin, Germany. [6]Deceased: Heiko Richter. ✉e-mail: heinz-wilhelm.huebers@dlr.de

Nov. 10, 11 and 13 2021. In total, 17 positions on Venus have been measured: seven on the dayside, nine on the nightside and one at the terminator. Atomic oxygen has been detected at all positions. We find column densities ranging from 0.7 to 3.8 ×10^17cm^{−2} between 15:00 and 21:00 LT with a maximum on the dayside where atomic oxygen is generated. The observed average Venus continuum brightness temperature is approximately 246 K corresponding to an altitude of about 65-70 km right above the cloud layer. The temperature of the atomic oxygen is approximately 156 K on the dayside and approximately 115 K on the nightside, which corresponds to altitudes around 100 km. These results are discussed in the context of other measurements of Venus' atmosphere.

## Results and discussion

The observation of atomic oxygen is based on the measurement of its ground state fine structure $^3P_1 \to {}^3P_2$ transition at 4.74 THz (63.2 μm)[14]. Its intensity depends on the temperature of the atmosphere as well as on the radiative properties and column density of the oxygen atoms. The transition has been detected in the mesosphere and thermosphere of Earth and Mars[15–19]. Recently, even the rare $^{18}O$ isotope has been identified by measuring this transition in the Earth atmosphere[20]. For Venus, Earth-bound absorption spectroscopy of near-infrared CO lines yields an isotope ratio that is not significantly different from that of the Earth[21], which is $^{16}O/^{18}O \approx 500$. In the atmosphere of Venus, the transition is expected to be thermally broadened with a line width of approximately 12 MHz full width at half maximum (FWHM) for emission originating from about 90 to 120 km altitude where the density of atomic oxygen is highest and the temperature is between 100 and 200 K[22]. The direct observation of this transition from the Earth is challenging, because of strong absorption by water vapor in the troposphere and lower stratosphere of Earth. Therefore, it requires airborne or spaceborne instruments.

The apparent diameter of Venus was approximately 29" and the phase angle approx. 42% (see methods, subsection "SOFIA/upGREAT observations"). Due to the geocentric velocity of Venus of about 13 km/s the atomic oxygen line is shifted by about 206 MHz from the telluric line. The high spectral resolving power of upGREAT allows for the ability to distinguish both. The telluric atomic oxygen line is used for frequency and radiometric calibration. A typical spectrum is shown in Fig. 1a. The atomic oxygen line originating from Venus is observed in absorption. In Fig. 1b an example of a dayside spectrum at the equator at 15:36 LT is shown. It should be noted that until now atomic oxygen has not been detected on the dayside of Venus. In contrast to nightglow observations from a Venus orbiter, which provide altitude profiles due to a limb-scan observing geometry, our method provides

column densities. However, with a THz heterodyne spectrometer on a Venus orbiter it would be possible to obtain altitude profiles of atomic oxygen.

In Fig. 1b the baseline corresponds to the continuum brightness temperature of Venus. The absorption signal at 4.744980 THz originates from higher levels in the Venus atmosphere and appears in absorption against the warm background of the lower atmosphere. It is offset by 203 MHz from the telluric signal at 4.744777 THz which has a rather peculiar shape. Due to the low elevation of the SOFIA telescope at the time of the observations, the telluric line is optically thick at the center and the corresponding temperature in the minimum is the brightness temperature of the Earth's lower thermosphere. In the wings, the telluric signal appears in emission, because this part originates from higher parts of the Earth's thermosphere where the temperatures are higher than in the Venus atmosphere and in the lower thermosphere (see methods, subsection "Continuum brightness temperature of Venus"). The telluric line is used to determine the brightness temperature of Venus. Since its central part is optically thick, the temperature equals the temperature of the Earth atmosphere (in this case 196 K). The atomic oxygen altitude distribution and the temperature profile of the Earth atmosphere, as provided by the semi-empirical NRLMSISE-00 (Naval Research Laboratory Mass Spectrometer and Incoherent Scatter Radar Exosphere – 2000) model (https://ccmc.gsfc.nasa.gov/modelweb/models/nrlmsise00.php), is taken as input for our radiative transfer model and the brightness temperature of Venus is a fit parameter (see methods, subsection "Continuum brightness temperature of Venus"). For the spectrum shown in Fig. 1b the brightness temperature of Venus corresponds to the baseline temperature of 245 K. Note that the minimum temperature of the telluric line is below the baseline, because it saturates at the temperature of the Earth atmosphere, which is lower than the brightness temperature of Venus.

The average continuum brightness temperature of all measured positions on Venus is 246 ± 6 K (Fig. 2a). This corresponds to temperatures at altitudes right above the cloud-top level at 65 and 70 km according to temperature profiles derived from observations at infrared and millimeter wavelengths[22–25]. For all LT the brightness temperatures of our measurements are in agreement with previous observations at similar wavelengths, which yield 255 ± 7 K in the wavelength region from 47 to 67 μm[26] and 240 K at 50 μm[24]. Sagawa[27] deduces a brightness temperature of 200 K at 60 μm. This rather low temperature has been attributed to pressure induced continuum absorption by $CO_2$ leading to an increased opacity of the atmosphere at this wavelength[27]. Our results do not confirm such an influence of $CO_2$. Closer inspection of the brightness temperature reveals a slight

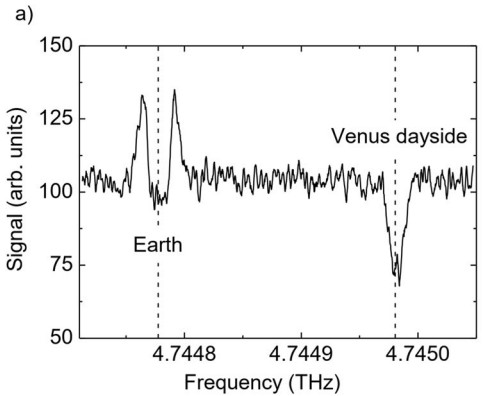

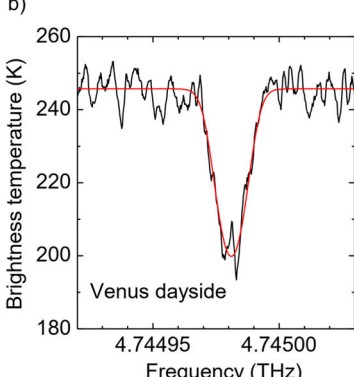

**Fig. 1 | Spectra of the fine structure $^3P_1 \to {}^3P_2$ transition of atomic oxygen in the atmospheres of Earth and Venus. a** Measured uncalibrated spectrum of atomic oxygen with the telluric line and the line originating from the dayside of Venus at 15:36 LT close to the equator (9.8° south). The Venus line is Doppler-shifted by 203 MHz from the telluric line. The centers of both lines are indicated by dashed lines. **b** Close-up of the Venus line in **a**. The temperature scale is calibrated. The red line is a fit with our radiative transfer model. Source data are provided as a Source Data file.

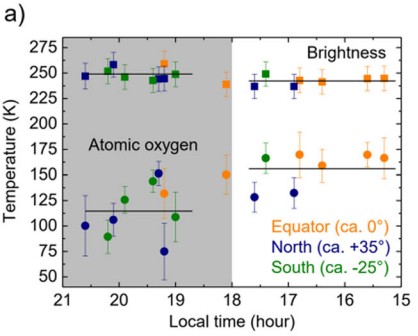
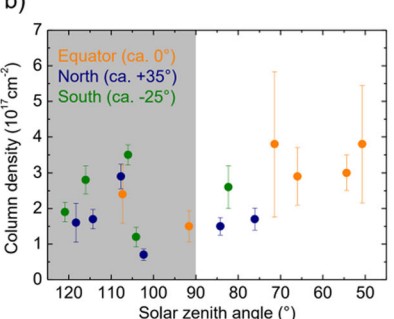
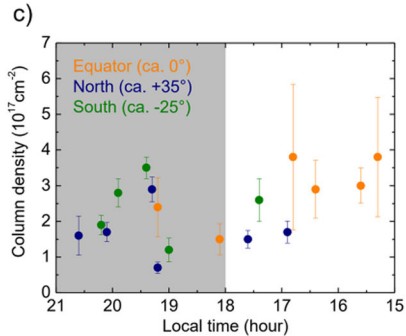

**Fig. 2 | Brightness temperature, atomic oxygen temperature and column density of Venus. a** Brightness temperatures and the temperatures of the atomic oxygen as a function of LT. The upper squared data points are the continuum brightness temperatures. The color code marks those pixels, which are close to the equator (approx. from -10° to +10°, orange), in the northern hemisphere around 35 ± 10° (blue) and in the southern hemisphere around −25° ± 10° (green). The lower data points (circles) are the temperatures of atomic oxygen. The grey area indicates nighttime. The straight lines are the average temperatures on the dayside and on the nightside. **b** Atomic oxygen column density as a function of the solar zenith angle. **c** Atomic oxygen column density as a function of LT (for **b**, **c** applies the same color code as in **a**). Note that sunset at the equator and at 100 km altitude occurs at 101.3° and 18:41 LT. The error bars represent 95% confidence intervals. Source data are provided as a Source Data file.

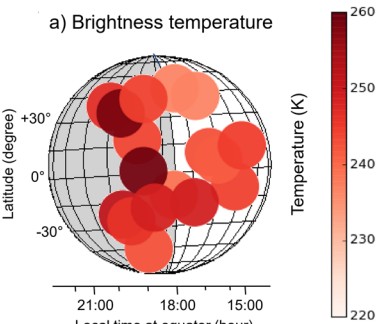
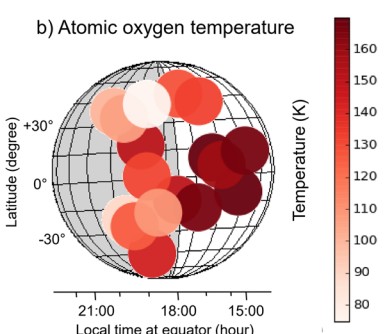
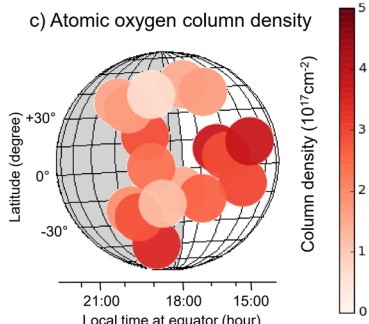

**Fig. 3 | Maps of temperature and atomic oxygen. a** brightness temperature, **b** atomic oxygen temperature, and **c** atomic oxygen column density of Venus. The light grey area of Venus marks the nightside. The evening terminator is the border between the white (daytime) and the grey (nighttime) area. The LT refers to the equator. The size of the circles corresponds to the FWHM of the telescope beam. Source data are provided as a Source Data file.

difference between nighttime (249 ± 6 K) and daytime (242 ± 4 K) (Fig. 2a). Although statistically barely significant, this could be attributed to a reduced opacity of the nightside atmosphere, i.e., the brightness temperature is determined by a lower atmospheric layer with slightly higher temperature. A map of the brightness temperature is shown in Fig. 3a.

In Fig. 2a the temperature of the atomic oxygen is shown for all positions on Venus as a function of LT. This is determined from the fit of a radiative transfer model (see methods, see subsections "Continuum brightness temperature of Venus" and "Column density and temperature of atomic oxygen"). The average temperature on the dayside is 156 ± 18 K. This is at the lower end of the dayside temperatures of 150 K to 200 K between 90 and 120 km reported by Limaye et al.[22]. It is also lower than the 200 K in the center of the line measured at 15:36 LT (see Fig. 1b) indicating that the line is not optically thick. The average nightside temperature (115 ± 25 K) is lower than the dayside temperature (156 ± 18 K). Also, the nightside temperature is at the lower end of the nightside temperatures of 110 K to 200 K[22]. For all LT no significant differences are observed for different latitudes (see also Fig. 3b). It is instructive to compare these temperatures with the FWHM of the atomic oxygen transition, which is thermally Doppler-broadened. The FWHM is 11.8 ± 0.9 MHz on the dayside and 10.2 ± 1.6 MHz on the nightside. The corresponding temperatures of 180 ± 25 K (dayside) and 135 ± 45 K (nightside) are in good agreement with the values from the radiative transfer model and further support that the temperature of atomic oxygen at the dayside is larger than at the nightside.

The absorption line originating from Venus is very well fitted by our radiative transfer model (Fig. 1b). The fit parameters of this model are the column density and the temperature of the atomic oxygen (see methods, see subsection "Column density and temperature of atomic oxygen"). The column density derived from this dayside spectrum at 15:40 LT is $(3.0 \pm 0.5) \times 10^{17}$ cm$^{-2}$. Figure 2b shows the column densities as a function of the solar zenith angle. At daytime with increasing solar zenith angle the column density decreases slightly, because the generation of atomic oxygen by photolysis of $CO_2$ decreases with decreasing illumination from the sun. At nighttime no trend of the column density is observable. Figure 2c shows the column densities for all pixels as a function of LT and Fig. 3c displays a map of the atomic oxygen column density across Venus. It varies between 0.7 and $3.8 \times 10^{17}$ cm$^{-2}$. Our values are similar to values obtained by models or derived from observations of the infrared $O_2$ nightglow and photochemical models[2,3]. Altitude profiles obtained from these nightglow data yield column densities between $3 \times 10^{17}$ cm$^{-2}$ and $6 \times 10^{17}$ cm$^{-2}$. The latter value is obtained at the antisolar point, where it reaches its maximum on the nightside and which is not covered by our observations. The column densities derived from a global circulation model and VIRTIS altitude profiles as reported in Brecht et al.[1] are lower (approximately $1 \times 10^{17}$ cm$^{-2}$). Also, results from a different model by Gilli et al.[5] yield column densities around $2 \times 10^{17}$ cm$^{-2}$, which are comparable with our observations. Daytime latitudinal variations predicted by the model of Gilli et al. are small, namely a decrease around 10% from the sub-solar point toward the terminator and towards high latitudes (50–70°)[5]. Our measurements indicate a small decrease of the

column density from about $(3\text{-}4) \times 10^{17}\,cm^{-3}$ at 15:00 to 17:00 h LT to $(1\text{-}2) \times 10^{17}\,cm^{-3}$ around the terminator. This might be caused by reduced photolysis as discussed above as well as recombination loss of atomic oxygen, which is transported from the dayside to the nightside. For comparison, Gérard et al. concluded that approximately half of the dayside atomic oxygen is transported to the nightside[28]. Between 19:00 and 20:00 LT a small local peak of the column density occurs. We speculate that this might be caused by dynamical processes in the atmosphere which may lead to a local maximum. This is, however, much less pronounced than the maximum at the antisolar point. It is worth noting that Soret et al. have identified a substructure of the antisolar atomic oxygen maximum from observations of the 1.27-μm nightglow of $O_2$ as well as highly variable, local $O_2$ maxima in regions close to the terminator[2,29]. We do not observe an increase of the column density towards the antisolar point, where the maximum column density is expected, because this starts at a later LT[1,2].

The direct measurement of the 4.7-THz fine structure transition of atomic oxygen opens an unexplored window for investigation of the Venus atmosphere at altitudes between the two major components of the global circulation in the region between the retrograde super-rotating zonal flow and the subsolar-to-antisolar flow, i.e. at about 90–120 km altitude where most of the atomic oxygen is concentrated. Atomic oxygen is generated on the dayside by photo-dissociation of $CO_2$ and CO. From there it is transported by the global circulation towards the anti-solar point. With increasing solar azimuth angle the generation of atomic oxygen decreases. In addition, a small fraction is lost by recombination. Therefore, its column density decreases near the terminator. Finally, as satellite measurements show, it accumulates at the converging stagnation point of the wind field near the antisolar point[1–3]. Future observations, especially near the antisolar and subsolar points but also at all solar zenith angles, will provide a more detailed picture of this peculiar region and support future space missions to Venus such as the DAVINCI (Deep Atmosphere Venus Investigation of Noble gases, Chemistry, and Imaging) mission of NASA[30] (National Air and Space Administration) or the EnVision mission of the European Space Agency (ESA)[31]. Along with measurements of atomic oxygen in the atmospheres of Earth and Mars these data may help to improve our understanding of how and why Venus and Earth atmospheres are so different.

## Methods

### SOFIA/upGREAT observations

The atomic oxygen fine structure line at 4.744777 THz was observed with the upGREAT[12] array heterodyne spectrometer, which was installed on SOFIA[13], a Boeing 747SP with a 2.5-m diameter telescope. The elevation of the diffraction-limited telescope beam can be varied from slightly below 20° to about 60°. Its azimuth is determined by the aircraft heading. The observations of Venus were made on Nov. 10, Nov. 11 and Nov. 13 2021 between 1:45 and 2:30 UTC (Coordinated Universal Time) with the telescope around 18.7° elevation. During observation the apparent diameter of Venus was approximately 29 arcsec and the phase was 42% (Table 1).

The 4.7-THz high frequency array (HFA) of upGREAT consists of seven hot-electron bolometer (HEB) mixers which are arranged in a hexagonal pattern around a central pixel[12,32]. The pixel separation is 13.6 arcsec and the FWHM of each beam is 6.3 arcsec. The local oscillator which pumps the array is based on a quantum-cascade

laser[33]. The single-sideband noise temperatures of the mixers vary between 2000 K and 4000 K depending on the pixel and the intermediate frequency which ranges from 0.5 to 4.0 GHz[34]. A digital fast Fourier transform spectrometer (FFTS) with a bandwidth of 4.0 GHz and a spectral resolution of 283 kHz serves as backend spectrometer[35]. Two blackbody radiators are used for calibration. One is at ambient temperature of $T_{hot} = 294\,K$ and the other is cooled to $T_{cold} = 149\,K$.

### Observing strategy

The nominal pointing of the SOFIA telescope, as defined by star tracker cameras, was on the brightness centroid of Venus. It is worth mentioning that these measurements are the first observation of a gaseous line in the Venusian atmosphere with SOFIA. Our observation took place during the same SOFIA flights as the unsuccessful attempt to detect phosphine in the Venusian atmosphere[36]. They were challenging, because Venus was close to the Sun and just above the horizon. This limited the observation time to about 20 min per flight and required on-time readiness of the complete observatory. The high sensitivity and spatial multiplexing of the upGREAT spectrometer has been crucial to map Venus in the short available time, against the strong attenuation of the signal by the atmosphere of the Earth due to the low elevation of the telescope. The distance between the outermost HEB pixels of the hexagonal array was 27.6 arcsec[32], which was about as large as the apparent diameter of Venus. In order to avoid having most of the pixels half on and half off Venus, the central pixel of the mixer array was pointed on the dayside at about 15:00 LT. In this case, two of the outer pixels were on the nightside while two other pixels were off Venus. The upper and lowermost pixels were on the rim of Venus. The array was oriented with one line of mixers parallel to the terminator of Venus. After completion of the measurements in this position it was rotated by 180° in order to shift the pixels by 5 arcsec, because of a 2.5 arcsec offset of the central HFA pixel with respect to the optical pointing. The nominal pointing was corrected by a few arcsec using the HFA pixels which are partly on and partly off Venus. The telluric lines of these pixels were fitted with an appropriate background temperature, which was derived from a convolution of the cold sky background and Venus. The pointing accuracy was 3.7 arcsec. Spectra were measured in double beam switching mode with a 0.6-Hz chopping frequency (synchronized with the cryocooler mechanical cycle time in order to minimize standing wave patterns introduced by the optical modulation of the path lengths within the instrument) and a throw of 90 arcsec with the whole array pointing either onto the cold sky (off-spectra) or onto Venus (on-spectra). In both cases the integration time was 13.8 s. For further analysis, all of the on-spectra for each pixel position were integrated. Note that, due to the short observation time for Venus, the number of integrated spectra varied for the positions on Venus and, therefore, their signal-to-noise ratio and the measurement uncertainty in Fig. 2 are different.

### Radiative transfer code

The radiative transfer code used for analysis of the spectra was originally developed for analysis of the atomic oxygen emission in the atmosphere of Earth[18]. Here, we apply the code to both, the emission from Earth and the emission from Venus. First, the radiance is analyzed

**Table 1 | Observing conditions of Venus**

| Date | Geocentric distance (au) | Apparent diameter (arcsec) | Phase angle (%) | Geocentric velocity (km/s) | Solar activity (Earth, F.10.7) |
|---|---|---|---|---|---|
| Nov. 10 | 0.58 | 28.7 | 43.2 | −13.06 | 80 |
| Nov. 11 | 0.57 | 29.0 | 42.6 | −13.03 | 80 |
| Nov. 13 | 0.56 | 29.8 | 41.4 | −12.97 | 80 |

with at least 20 spectral bins across the line according to

$$R_\nu = \int_z B_\nu(T(z)) \frac{\partial \tau_\nu}{\partial z} dz, \qquad (1)$$

with $R_\nu$ being the radiance at frequency $\nu$. $T(z)$ is the temperature and $B_\nu(T(z))$ is the Planck blackbody function and source function at altitude $z$. The monochromatic transmission $\tau_\nu$ at frequency $\nu$ is given by

$$\tau_\nu = \exp(-\sigma_\nu u). \qquad (2)$$

Here, $\sigma_\nu$ is the absorption cross section given by the product of the line strength $S$ and the line shape of the atomic oxygen transition and $u$ is the optical mass along the line of sight. In the model scattering is neglected, i.e., the emissivity is given by $\varepsilon_\nu = 1 - \tau_\nu$. Pressure broadening can be neglected because of the low pressure in both atmospheres at altitudes where atomic oxygen prevails and the line is as good as purely Doppler broadened. The line strength, $S$, depends on the temperature $T$. $S = 1.131 \times 10^{-21}$ cm$^{-1}$ atom$^{-1}$ cm$^2$ at $T_{ref} = 296$ K. It can be calculated for other temperatures according to

$$S(T) = S(T_{ref}) \frac{Q(T_{ref})}{Q(T)} \frac{\exp\left(\frac{-E_L}{kT}\right)}{\exp\left(\frac{-E_L}{kT_{ref}}\right)} \frac{1 - \exp\left(\frac{-h\nu}{kT}\right)}{1 - \exp\left(\frac{-h\nu}{kT_{ref}}\right)}. \qquad (3)$$

Here, $Q$ is the electronic partition function and $E_L$ is the energy of the lower state of the transition, which is zero because $^3P_2$ is the ground state. The radiative transfer code evaluates the radiance in units of temperature across the atomic oxygen line with a step size of 244 kHz, i.e., matching the spectral resolution of the FFTS. The total radiance is determined by integrating over the entire line. The code was validated by comparing the radiances obtained with our code with those reported in Mlynczak et al[17]. using the concentration profiles and temperature profiles provided by NRLMSISE-00 (https://ccmc.gsfc.nasa.gov/modelweb/models/nrlmsise00.php). Our calculations agree within 1% with the radiances calculated by Mlynczak et al.[17], which in turn agree within 1% with other radiative transfer codes, namely FUTBOLIN (Full Transfer by Ordinary Line-by-Line Methods) and MRTA (Monochromatic Radiative Transfer Algorithm)[37,38]. Throughout the analysis, local thermodynamic equilibrium is assumed to prevail in both atmospheres. For more details see Richter et al[18].

### Continuum brightness temperature of Venus

The continuum brightness temperature of Venus is determined by analyzing the telluric atomic oxygen line in the off- and on-spectra. Due to the low telescope elevation and the long path through the atmosphere the telluric line is saturated in both spectra. When measured against the cold background of the sky, the line appears in emission while when measured against Venus the line appears in the center in absorption and at the wings in emission (Fig. 4). In both cases, saturation in the line center occurs at the same temperature, which is the temperature of the Earth atmosphere at the altitude where the line becomes optically thick. In this case, it corresponds to the lower

thermosphere, at around 100 km altitude. The emission in the line wings is due to the fact that at lower opacities the telluric absorption is dominated by higher and warmer layers in Earth's thermosphere.

Using the radiative transfer model described above, the temperature of the Earth atmosphere at which the line saturates is determined. For these calculations the atmosphere of the Earth between 400 km and 50 km is considered and divided into 1 km thick, spherical layers. Scattering is neglecting. For each layer, the transmission $\tau_\nu$ is calculated according to Eq. (3). with the atmospheric temperature and atomic oxygen density of Earth provided by NRLMSISE-00 for the flight dates and geolocation of our measurements. According to NRLMSISE-00 the atmospheric temperature as well as the atomic oxygen density do not change during the short Venus observation time. With a least-squares procedure the telluric line is fitted with the temperature of the cold sky (2.7 K) as source function. This fit (straight blue line in Fig. 4) provides the maximum at the line center which corresponds to the temperature as provided by NRLMSISE-00 (in this case 196 K) and provides scaling factor for the spectra.

For further analysis the shape of the telluric line in the on-spectra is fitted with the brightness temperature of Venus as fitting parameter. The temperature at the minimum of the telluric line is the temperature at saturation, namely 196 K. The coupling efficiency of the telescope is taken into account by a convolution of the telescope beam with the solid angle of Venus. Losses which are caused by the telescope's central blockage and spillover are taken into account by a coupling efficiency of 0.8. By this means the brightness temperature is determined for each pixel. The uncertainty is determined as the Gaussian sum of the uncertainties of the coupling efficiency (±0.03) and the fitting procedure.

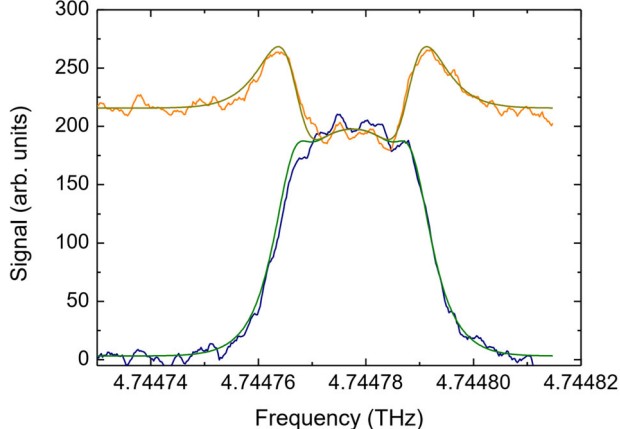

**Fig. 4 | Atomic oxygen line originating from the Earth's atmosphere.** When observed against the cold sky the line appears in emission (blue) and when observed against Venus it appears in absorption at the center and in emission at the wings (orange). Note that in the center of the line, its intensity is the same independently whether it is observed in emission or in absorption. The straight lines are radiative transfer calculation with the cold sky (green) or Venus (dark yellow) as background brightness temperature. Source data are provided as a Source Data file.

### Table 2 | Data around the equator

| Latitude (deg) | Longitude (deg) | Local time (h:min) | Solar zenith angle (deg) | Brightness temperature (K) | Atomic oxygen temperature (K) | Column density (10$^{17}$ cm$^{-2}$) |
|---|---|---|---|---|---|---|
| 11.6 | 295.1 | 15:18 | 50.7 | 244.8 | 166.5 | 3.8 |
| −9.8 | 282.6 | 15:36 | 54.4 | 244.5 | 169.7 | 3.0 |
| 5.0 | 273.0 | 16:24 | 66.0 | 241.3 | 159.0 | 2.9 |
| 9.4 | 267.6 | 16:48 | 71.4 | 242.8 | 169.8 | 3.8 |
| −12.3 | 244.8 | 18:06 | 91.6 | 238.9 | 150.2 | 1.5 |
| 1.2 | 231.7 | 19:12 | 107.3 | 259.0 | 131.9 | 2.4 |

## Table 3 | Data of the northern hemisphere

| Latitude (deg) | Longitude (deg) | Local time (h:min) | Solar zenith angle (deg) | Brightness temperature (K) | Atomic oxygen temperature (K) | Column density ($10^{17}$ cm$^{-2}$) |
|---|---|---|---|---|---|---|
| 41.6 | 272.3 | 16:54 | 76.1 | 236.8 | 132.4 | 1.7 |
| 47.9 | 252.6 | 17:36 | 84.2 | 236.8 | 128.3 | 1.5 |
| 41.6 | 237.1 | 19:12 | 102.3 | 244.5 | 74.9 | 0.7 |
| 17.9 | 226.7 | 19:18 | 107.7 | 244.1 | 151.5 | 2.9 |
| 33.6 | 218.0 | 20:06 | 114.3 | 258.2 | 106.1 | 1.7 |
| 38.3 | 210.1 | 20:36 | 118.3 | 246.9 | 100.2 | 1.6 |

## Table 4 | Data of the southern hemisphere

| Latitude (deg) | Longitude (deg) | Local time (h:min) | Solar zenith angle (deg) | Brightness temperature (K) | Atomic oxygen temperature (K) | Column density ($10^{17}$ cm$^{-2}$) |
|---|---|---|---|---|---|---|
| −17.5 | 263.9 | 17:24 | 82.3 | 249.0 | 166.4 | 2.6 |
| −18.9 | 240.9 | 19:00 | 104.1 | 248.8 | 108.8 | 1.2 |
| −45.6 | 224.8 | 19:24 | 106.0 | 242.7 | 143.9 | 3.5 |
| −26.3 | 220.8 | 19:54 | 116.0 | 246.0 | 125.7 | 2.8 |
| −21.8 | 216.3 | 20:12 | 120.9 | 251.8 | 89.4 | 1.9 |

### Column density and temperature of atomic oxygen

For determination of the column density of the atomic oxygen the radiative transfer code is used. In contrast to the analysis of the telluric line the Venus atmosphere is modeled as a single layer with a constant temperature and constant atomic oxygen density. In this case, the source function equals the Planck function for the temperature of the atomic oxygen layer and the brightness temperature of Venus is taken as input for the source function. This simplification is possible because the atomic oxygen is concentrated in a rather small altitude range between 90 and 120 km where the atmospheric temperature does not change much[22]. With the Venus brightness temperature as input, this simplified radiative transfer model returns the temperature and the column density of the atomic oxygen. The uncertainty of the temperature and the column density is determined from the fit and the uncertainty of the continuum brightness temperature and represents a 95% confidence interval. The data resulting from this analysis are given in Tables 2–4.

### Data availability

The data for the atomic oxygen emission profiles generated in this study have been deposited in the SOFIA data archive under accession code https://irsa.ipac.caltech.edu/applications/sofia/?_action=layout.showDropDown&view=Search. Search terms are: "Spatial Constraints: Solar System Target, Object Name: Venus", "Proposal Constraints: Primary Investigator: Heinz-Wilhelm Huebers, Plan ID: 75_0068", "Observation Constraints: Observation Date: 2021-11-10 to 2021-11-13", "Instrument Constraints: GREAT", "Data Product Constraints: Level 1". Source data are provided with this paper.

### Code availability

The code for calculating the NRLMSISE-00 profiles can be accessed at https://ccmc.gsfc.nasa.gov/modelweb/models/nrlmsise00.php. The radiative transfer code is available from the authors upon request.

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

## Acknowledgements

upGREAT is a development by the Max-Planck-Institut (MPI) für Radioastronomie and the Kölner Observatorium für SubMillimeter Astronomie (KOSMA)/Universität zu Köln, in cooperation with the Deutsches Zentrum für Luft- und Raumfahrt (DLR; German Aerospace Center) Institut für Optische Sensorsysteme. The development of upGREAT is financed by the participating institutes, by the German Aerospace Center (DLR) under grants 50 OK 1102, 1103, and 1104, and within the Collaborative Research Centre 956, funded by the Deutsche Forschungsgemeinschaft (DFG). SOFIA was jointly operated by the Universities Space Research Association (USRA), under NASA contract NAS2-97001, and the Deutsches SOFIA Institut (DSI), under DLR contracts 50 OK 0901 and 50 OK 1301 to the University of Stuttgart. We thank the SOFIA operations and engineering teams for their dedication and supportive responses to our requests, enabling the first observation of Venus with SOFIA. A personal note from the authors: While writing the revision of the manuscript our co-author Heiko Richter passed away. He was an excellent scientist, an esteemed colleague, and a good friend. We will always remember him.

## Author contributions

H.-W.H. conceived the idea and proposed the observation. J.S. planned the observations and analyzed the pointing model offsets in the data. H.R. and H.-W.H. analyzed and interpreted the data with contributions from U.U.G., R.G., B.K., and H.W. H.-W.H. wrote the text with contributions from all authors.

## Funding

## Competing interests

The authors declare no competing interests.
