## [Peer Review File · Nature Communications]

REVIEWER COMMENTS

Reviewer #1 (Remarks to the Author):

The manuscript « Atomic oxygen on the dayside and nightside of Venus » by Hübers et al. presents a method applied to Venus for the first time to determine oxygen column densities and temperatures.

The technique, based on the observation of the 63 μm oxygen emission wavelength using an airborne spectrometer is described. This method leads to the measurement of the Earth atmosphere temperature, as well as the Venus cloud top temperature, the atomic oxygen temperature and the atomic oxygen column density. This is the first time that such direct measurements are performed to estimate the oxygen column density on the Venus dayside. The method is also valid for the Venus nightside. Results are in good agreement with previous studies, based on airglow measurements and global circulation models.

The SOFIA/upGREAT observations have been acquired between 15 and 21 LT. Future observations are very promising to map the entire Venus globe, especially near both the sub-solar and the anti-solar points, which are missing in this study.

The paper is clear, well written and results are highly encouraging. I recommend this paper to be published by Nature Communications after addressing the following points.

Main comments

L. 31: “The atomic oxygen is found at altitudes around 100 km”: this study only provides column densities. It is thus not possible to retrieve altitude of the oxygen layer from those measurements. This statement is not wrong but based on results from previous studies. In the abstract, the statement is misleading, as the reader might think that this result comes from this study.

L.62: please, mention the limitation of your method, that only provides column densities, while limb observations of the airglow can lead to atomic oxygen density profiles. These methods are thus complementary.

l.66: you mention the detection of the 63 μm emission in the Earth and Mars atmospheres. Is it expected to be seen elsewhere? Also, please, mention earlier studies for the Earth detections, such as Grossman and Vollmann (1997), Grossman et al. (2000) and Mlynczak et al. (2004).

l.67: you mention the ^{18}O isotope on Earth. Is it expected to be observed on Venus? If yes, please, mention it and, if not, maybe it is not useful to write this sentence in this manuscript.

l.81: It would be great if you could provide a local-time/latitude map of the locations of the observations on the planet. Or maybe a table with the latitude, longitude, local time and SZA of the 17 observations. It could also be inserted in the methods section.

Figure 1a: please, add a vertical dashed line at 4.744777 THz and another one at 4.744980 THz and explain in the legend.

Figure 2a and 2b: for clarity, a vertical dashed line should be added at 18:00 LT to mark the terminator. Or maybe the background of the left part of the plots should be grey (for consistency with figure 3).

Figure 2a: I am not sure I understand. Why don't we see the atomic oxygen temperature of 200 K at 15:40 LT shown in figure 1b?

Figure 2b: in figure 2a, you use squares for the brightness temperatures and circles for the atomic oxygen temperatures. Please, use different symbols for figure 2b, which represent another quantity: oxygen densities (or, at least, replace squares by circles).

According to the SS-AS circulation theory, and as explained in your conclusion, the atomic oxygen density is expected to be maximum at the subsolar point, then decrease near the terminator and increase again at the anti-solar point. Yet, this is not exactly what we can see in Figure 2b. The decrease from 15:00 to 18:00 LT can clearly be observed, which is expected. But why is the oxygen density increasing again between 18:00 and 19:00 LT, and then decreasing from 19:00 to 21:00? We should expect some symmetry of the dayside and the nightside points but we do not see that on this plot. Do you have an explanation for this behavior in the nightside?

I also suggest that you try to plot these figures according to the distance to the subsolar or antisolar points, instead of latitudes. Soret et al. (2012) showed that, since the distribution is expected to be concentrically distributed around the antisolar point on the nightside, data should be represented as a function of the antisolar angle (figure 2 of their manuscript). Maybe a trend will more clearly appear in your dataset if using this variable.

It is also a shame that no data have been acquired at the SS and AS points. I hope future observations will be able to fill in the gaps. That would be a huge step in the understanding of the Venus circulation.

L.157: please, mention that the $6 \times 10^{17} \text{ cm}^{-2}$ value is obtained at the antisolar point, where it reaches its maximum on the nightside, while your dataset does not go so deep in the nightside.

L.160-164: Again, your data set does not include observations at the antisolar and subsolar points. Being mostly performed near the terminator, the small variations retrieved in this study can (and should) easily be explained.

L.160: why don't you mention the work of Gilli sooner?

L.176: I agree with your explanation from a theoretical point of view. But, as explained in my comments of figure 2, I do not agree that you see this trend in your data (except maybe on the dayside, but you are missing the 12:00 to 15:00 LT area anyway). You should rephrase this paragraph.

L.184: This study can clearly show how Venus and Earth are different, but it will not be straightforward, using the oxygen density, to understand how they evolved so differently. I think you should rephrase.

I.252 and 283: “enabling the first observation of Venus with SOFIA” Very good! It should appear in the main text.

Figure M1: It is not easy to distinguish between the data and the fits. Could you maybe lighten the fit curves or use different colors?

I.385: “the atmospheric temperature does not change much”. Please, add a reference.

Overall, this is a very interesting work, with promising results and I wonder whether if you have considered studying the correlation between your atomic oxygen column density and the O₂ nightglow observed by Venus Express? Or, to compare your atomic oxygen temperature and the O₂ nightglow? These are two observational datasets that do not require any modeling and it would definitely be a good way to connect them and bring this work a step further.

Minor comments

I.23: I suggest you add “However, past and current detection methods”.

I.27: “on board SOFIA”.

L.26-27: Maybe the upGREAT and SOFIA acronym should be explained in the abstract.

I.28: please, add “0.7 to 3.8 x 10¹⁷ cm⁻² between 15:00 and 21:00 Local Time”.

I.32: “These data are the first dayside measurements of atomic oxygen.” It should appear sooner in the abstract. Maybe, I.25 “We report on the first direct detection...”.

I.32: “This new method”

I.39-40: please, add commas “On the nightside, atomic oxygen recombines in a three-body reaction to molecular oxygen. This is the main source of excited oxygen molecules, which, in turn, are the source of the Venus nightglow.”

I.43: what does “This” refer to? The global circulation?

I.48: “limb” Do you mean terminator?

I.51: “onboard” instead of “on”

L.52: please, provide the transition of the 1.27 μm airglow.

I.54: add comma “quenching coefficients, reaction rates and efficiencies, atomic oxygen densities have been”

L.55: add reference 3 together with reference 2

I.57: change “longitude” to “latitude”

l.59: “stronger”: please, quantify.

l.68: add a comma after “In the atmosphere of Venus”

l.74-75: the acronym should probably be explained in the abstract.

l.75: “on the SOFIA airplane”. And, also, add a reference to Methods/SOFIA upGREAT observations.

l.80: add a comma after “In total”.

l.79-80: the sentence is a bit long. I suggest “high spectral resolving power of upGREAT allows distinguishing both. The telluric atomic oxygen line is used for frequency and radiometric calibration.”

l.81: replace the comma after “measured” by “:”.

l.107: please, change to 246 K, for consistency with the abstract and the rest of the manuscript.

l.114-115: replace “which are on” by “in”.

l.115: a space should be added between “points” and “(circles)”.

l.123-124: avoid the repetition of the word “measurements” in the same sentence.

l.125: “predicts” sounds like the result of a model, while Sagawa (2008) corresponds to an observational study. Maybe “observes” or “deduces” are more appropriate terms here.

l.136: “This is determined from the fit of a radiative transfer model”.

l.138: change “ref 16” to the names of the authors, as done in the rest of the manuscript.

L.158: the reference “2” to Brecht et al is a little bit weird at its current location. It looks like a squared value.

Figure 3: For clarity, it would be nice to add more clearly some local time values (12:00, 18:00 and 24:00 LT, at least).

l.176: use a period instead of a colon.

l.179: replace “towards the nightside” by “near the terminator”.

l.130: “Future observations, especially near the antisolar and subsolar points but also at all solar zenith angles, will provide...”

l.227: ref 17 is not cited in the text.

l.282: please, add commas “The nominal pointing of the SOFIA telescope, as defined by star tracker cameras, was ...”

Please, add commas after “half on Venus” (l. 291), “For further analysis” (L.305), “Note that” (l.306), “for Venus” (l. 306), “Throughout the analysis” (l. 313), “Here” (L. 325), “the sky” (l.343), “In both cases” (l.344), “center of the line” (l.354), “above” (L.358), “For each layer” (l.361), “input” (l.386).

L.312: please, add a reference to the Earth's study.

L.317: I think it should be equation (1) instead of (2). Same for the following equations.

L.329: $S=1.131 \times 10^{-21}$ (x is missing)

L.346: the sentence is a bit long. I suggest "... Earth atmosphere at the altitude where the line becomes optically thick. In this case, it corresponds to the lower thermosphere, at around 100 km altitude."

Reviewer #2 (Remarks to the Author):

Overall:

The manuscript is discussing a unique dataset that was obtained by the upGREAT heterodyne spectrometer on board SOFIA. Atomic oxygen is an important chemical species in atmospheres but can also be difficult to directly observe. This manuscript is providing a decent and brief overview of the importance of atomic oxygen in the Venusian atmosphere. It also does a good job in comparing their results to past observations and model simulations. There are a few revisions being suggested that are easily corrected. The recommendation is this manuscript needs minor revisions before publication.

Major comments:

(1) The introduction/motivation section (~Lines 38 – 44) needs to be re-organized and focused. Things are briefly touched on but the connection between the points are not clear or fully covered with appropriate references. It is unclear if the authors want to only discuss one nightglow (O₂ IR) or briefly state a few.

Atomic oxygen is important for photochemistry, circulation, and energy budget. Due to a lack of direct atomic oxygen observations, scientists have had to rely on other atmospheric features to understand the atomic oxygen distribution, such as nightglow. Nightglows (e.g. O₂ IR, O₂ visible, NO UV) provide insight into the photochemistry and dynamics due to the intensity and location of the features on the nightside

[e.g. Alexander et al. 1993 (<https://doi.org/10.1029/93JE00538>);

Bougher et al. 1990 (<https://doi.org/10.1029/JA095iA05p06271>);

Bougher et al. 1994 (<https://doi.org/10.1029/93JE03431>);

Garcia Munoz et al. 2009 (<https://doi.org/10.1029/2009JE003447>);

Brecht et al. 2011 (<https://doi.org/10.1029/2010JE003770>);

Gerard et al. 2017 (<https://doi.org/10.1007/s11214-017-0422-0>);

Navarro et al. 2021 (<https://doi.org/10.1016/j.icarus.2021.114400>)].

Understanding atomic O distribution is greatly important for CO₂ 15 micron cooling (the dominate cooling mechanism in the upper atmosphere), thus important for energy balance. [e.g. Bougher et al. 1994 (<https://doi.org/10.1029/94JE01088>);

Gilli et al. 2021 (<https://doi.org/10.1016/j.icarus.2021.114432>)].

(2) Throughout the manuscript, please be clearer on the local time being discussed. There are places where it isn't clear if the full range of local times are being discussed, the dayside, or the nightside are being discussed.

(3) The radiative transfer (RT) code needs more support. It is unclear if the RT code has been published before and how the input values were determined. This section needs references and/or more detail.

Details:

Line 20: replace “is mostly located” with “peaks”

Line 41: replace “source of the Venus nightglow” with “source of several Venus nightglows”. O₂ IR nightglow and the O₂ Visible nightglow can be produced from the 3-body reaction.

Line 39 – 44: This group of sentences needs rephrasing and re-organization. See major comment #1

Line 44: “Pioneer” -> “Pioneer Venus Orbiter”

Line 50: State the wavelength or just “IR” after “O₂”

Line 53-55: This sentence is missing a word or punctuation to provide more clarity in the statement “reaction rates and efficiencies atomic oxygen densities have been calculated.”

Line 54: Is temperature a-priori information?

Line 79: it is unclear what is meant by “allows distinguishing both and using the telluric...”

Line 138: check on how to write reference within statement

Line 140: “The average nightside temperature is lower” ... what is it lower than?

Line 145: remove parentheses from around the temperature values.

Line 151: Please clarify the column density value stated on line 152 is for the dayside.

Line 154: Please clarify “These values”. Is this regarding the range or the average or all of the above?

Line 155: "O2 nightglow", do you mean "O2 IR nightglow"?

Line 156: Please clarify "these data". It is unclear if this refers to the new observations or from previous work. Also unclear which local time this refers to.

Line 157: A clearer statement might be: "The column densities derived from VIRTIS altitude profiles as reported in Brecht et al. 2012 are lower ($\sim 1 \times 10^{17} \text{ cm}^{-2}$)."2."

Line 157-158: Brecht et al. 2012 report GCM simulation results too. Should state those values to compare with the VIRTIS results, your results, and with the next sentence about a different GCM simulation result.

Line 158: missing period at end of sentence

Line 159: rephrase; "Also, results from a different model yielded column densities around $2 \times 10^{17} \text{ cm}^{-2}$ which are comparable with our observations."

Line 168 – 170: (Fig. 3 Caption) Please state if this is the evening or morning terminator

Line 283-284: Please clarify this statement. There are other SOFIA observations of Venus. Most of them are conference proceedings, but Cordiner et al. 2022 (<https://doi.org/10.1029/2022GL101055>) is actually published.

Line 291: Please clarify or rephrase; "In order to avoid that most of the pixels..." What is being avoided?

Line 305: a comma is needed after, "For further analysis"

Line 311: Is there a reference for the radiative transfer code? If the code is not public and/or published, please add supporting references for values etc.

Line 358: Please rephrase this sentence. What temperature is determined?

Line 396 – 399: Could you please provide more information, such as key words to put in the search fields to find the exact data that was used? As written, it is unclear if the data is available.

Berlin, July 22, 2023

Dear Reviewers:

Thank you for your thoughtful comments. We have included all of them in the revised manuscript. In the red-line version of our manuscript (Huebers-Venus-revision-red-line-2023-07-12) these changes are indicated in red. On the following pages you will find detailed comments on your remarks (in red). We hope that the revised manuscript meets your expectations.

Sincerely,

Heinz-Wilhelm Hübers (on behalf of the authors)

REVIEWER COMMENTS

Reviewer #1 (Remarks to the Author):

The manuscript « Atomic oxygen on the dayside and nightside of Venus » by Hübers et al. presents a method applied to Venus for the first time to determine oxygen column densities and temperatures.

The technique, based on the observation of the 63 μm oxygen emission wavelength using an airborne spectrometer is described. This method leads to the measurement of the Earth atmosphere temperature, as well as the Venus cloud top temperature, the atomic oxygen temperature and the atomic oxygen column density. This is the first time that such direct measurements are performed to estimate the oxygen column density on the Venus dayside. The method is also valid for the Venus nightside. Results are in good agreement with previous studies, based on airglow measurements and global circulation models.

The SOFIA/upGREAT observations have been acquired between 15 and 21 LT. Future observations are very promising to map the entire Venus globe, especially near both the sub-solar and the anti-solar points, which are missing in this study.

The paper is clear, well written and results are highly encouraging. I recommend this paper to be published by Nature Communications after addressing the following points.

Main comments

L. 31: “The atomic oxygen is found at altitudes around 100 km”: this study only provides column densities. It is thus not possible to retrieve altitude of the oxygen layer from those measurements. This statement is not wrong but based on results from previous studies. In the abstract, the statement is misleading, as the reader might think that this result comes from this study.

Our study provides the temperature of the atomic oxygen and the column density. With a-priori knowledge of the temperature profile of the Venusian atmosphere it is, therefore, possible to obtain coarse altitude information, but not altitude profiles as it is possible with limb-scanning instruments on Venus orbiters. We have rephrased the sentence. Now it reads: “The temperature of the atomic oxygen is ~ 156 K on the dayside and ~ 115 K on the nightside which corresponds to altitudes around 100 km.” (lines 32-34). With this

reformulation we hope to clarify that we determine the temperature of atomic oxygen which corresponds to an altitude.

L.62: please, mention the limitation of your method, that only provides column densities, while limb observations of the airglow can lead to atomic oxygen density profiles. These methods are thus complementary.

Our method cannot deliver altitude profiles, because they are from an aircraft which does not allow limb observations. If a THz heterodyne spectrometer would be implemented on a Venus orbiter, it would also be possible to derive altitude profiles with our method. We have added the following two sentences (lines 89 - 92) to explain this: "In contrast to nightglow observations from a Venus orbiter, which provide altitude profiles due to a limb-scan observing geometry, our method provides column densities. However, with a THz heterodyne spectrometer on a Venus orbiter it would be possible to obtain altitude profiles of atomic oxygen."

I.66: you mention the detection of the 63 μm emission in the Earth and Mars atmospheres. Is it expected to be seen elsewhere? Also, please, mention earlier studies for the Earth detections, such as Grossman and Vollmann (1997), Grossman et al. (2000) and Mlynczak et al. (2004).

Atomic oxygen has been detected, for example, in the atmosphere of Europa (D. T. Hall et al., Nature, vol. 373, pp. 677-679, 1995). But these observations are in the UV. The only planets (incl. moons) where the 4.7-THz transition was detected are Earth, Mars and Venus. In principle, the 4.7-THz line might be detected in the atmosphere of Europa, but this is challenging, because the atomic oxygen column density is three orders of magnitude smaller than on Venus. Regarding Earth: We have added the suggested references.

I.67: you mention the ^{18}O isotope on Earth. Is it expected to be observed on Venus? If yes, please, mention it and, if not, maybe it is not useful to write this sentence in this manuscript.

We have added the following sentence about the ^{18}O isotope in the Venusian atmosphere (lines 71 - 73): "For Venus, Earth-bound absorption spectroscopy of near-infrared CO lines yielded an isotopic ratio that is not significantly different from that of the Earth with $^{16}\text{O}/^{18}\text{O} \approx 500$ (Iwagami et al., Ref. 21)."

I.81: It would be great if you could provide a local-time/latitude map of the locations of the observations on the planet. Or maybe a table with the latitude, longitude, local time and SZA of the 17 observations. It could also be inserted in the methods section.

Fig. 3 is a local-time/latitude map. But numbers of the exact positions are not given. We have added local times and latitudes as well as a table with the requested information in the methods section (Tables 2-4).

Figure 1a: please, add a vertical dashed line at 4.744777 THz and another one at 4.744980 THz and explain in the legend.

We have added the vertical lines for both frequencies and explained it in the figure caption.

Figure 2a and 2b: for clarity, a vertical dashed line should be added at 18:00 LT to mark the terminator. Or maybe the background of the left part of the plots should be grey (for consistency with figure 3).

We changed it according to the recommendation. The background of the nightside data is grey.

Figure 2a: I am not sure I understand. Why don't we see the atomic oxygen temperature of 200 K at 15:40 LT shown in figure 1b?

The atomic oxygen line is not optically thick and not optically thin, but somewhere in between. If the line would be optically thick we would see the temperature of atomic oxygen, which is 170 K at 15:40 LT. This is explained in lines 115 – 117.

Figure 2b: in figure 2a, you use squares for the brightness temperatures and circles for the atomic oxygen temperatures. Please, use different symbols for figure 2b, which represent another quantity: oxygen densities (or, at least, replace squares by circles).

We have changed it. Now, the old Fig. 2b is Fig. 2c and circles represent the oxygen densities.

According to the SS-AS circulation theory, and as explained in your conclusion, the atomic oxygen density is expected to be maximum at the subsolar point, then decrease near the terminator and increase again at the anti-solar point. Yet, this is not exactly what we can see in Figure 2b. The decrease from 15:00 to 18:00 LT can clearly be observed, which is expected. But why is the oxygen density increasing again between 18:00 and 19:00 LT, and then decreasing from 19:00 to 21:00? We should expect some symmetry of the dayside and the nightside points but we do not see that on this plot. Do you have an explanation for this behavior in the nightside?

We speculate that the increase from 19:00 to 21:00 is a local maximum of atomic which might be induced by the local wind field structure oxygen (for example caused by a vortex). It seems that this happens in a region covering the on the northern (two data points with small uncertainty) and southern hemisphere (one data point with small uncertainty) as well as the equator (although the one data point has a rather large uncertainty). It is worth noting, that some substructure is also observed by VIRTIS for the atomic oxygen concentration around the antisolar point (Soret et al., 2012, Ref. 3). However, we have only few data points and therefore it is not possible to make a final conclusion about that. We have added a discussion on this topic (lines 187 – 191) which reads: "Between 19:00 and 20:00 LT a small local peak of the column density occurs. We speculate that this might be caused by dynamical processes in the atmosphere which may lead to a local maximum. This is, however, much less pronounced than the maximum at the antisolar point. It is worth noting, that Soret et al. have observed a substructure of the antisolar atomic oxygen maximum with local maxima³."

I also suggest that you try to plot these figures according to the distance to the subsolar or antisolar points, instead of latitudes. Soret et al. (2012) showed that, since the distribution is expected to be concentrically distributed around the antisolar point on the nightside, data should be represented as a function of the antisolar angle (figure 2 of their manuscript). Maybe a trend will more clearly appear in your dataset if using this variable. It is also a shame that no data have been acquired at the SS and AS points. I hope future observations will be able to fill in the gaps. That would be a huge step in the understanding of the Venus circulation.

Thank you for the suggestion. We have included a plot (new Fig. 2b) with the column density as a function of the SZA. The nighttime values have been averaged (weighted average with the uncertainty as weight). The trend of decreasing column density with increasing SZA is visible. We have discussed this in lines 166 – 170, which reads: “Fig. 2b shows the column densities as a function of the solar zenith angle. The black data point at nighttime is the weighted average of all nighttime column densities ($(1.67 \pm 0.09) \times 10^{17} \text{ cm}^{-2}$). With increasing solar zenith angle the column density decreases slightly, because the generation of atomic oxygen by photolysis of CO_2 decreases with decreasing illumination from the sun.”

I.157: please, mention that the $6 \times 10^{17} \text{ cm}^{-2}$ value is obtained at the antisolar point, where it reaches its maximum on the nightside, while your dataset does not go so deep in the nightside.

We have added the following sentence: “The latter value is obtained at the antisolar point, where it reaches its maximum on the nightside and which is not covered by our observations.” (lines 175 – 177).

I.160-164: Again, your data set does not include observations at the antisolar and subsolar points. Being mostly performed near the terminator, the small variations retrieved in this study can (and should) easily be explained.

We have added a plot with the column densities as a function of solar zenith angle (new Fig. 2b) and we discussed Fig. 2 a-c in more detail in lines 163-193.

L.160: why don't you mention the work of Gilli sooner?

We have rephrased the introductory parts (as suggested by reviewer 2) and added the reference to the work by Gilli et al. in the introduction (line 45).

I.176: I agree with your explanation from a theoretical point of view. But, as explained in my comments of figure 2, I do not agree that you see this trend in your data (except maybe on the dayside, but you are missing the 12:00 to 15:00 LT area anyway). You should rephrase this paragraph.

We have added Fig. 2b (column density vs. solar zenith angle) and discussed the Figure in more detail (lines 163 – 193).

L.184: This study can clearly show how Venus and Earth are different, but it will not be straightforward, using the oxygen density, to understand how they evolved so differently. I think you should rephrase.

We have rephrased that sentence more tentatively. It now reads “... data may help to improve our understanding of how and why Venus and Earth atmospheres are so different.”

I.252 and 283: “enabling the first observation of Venus with SOFIA” Very good! It should appear in the main text. Figure M1: It is not easy to distinguish between the data and the fits. Could you maybe lighten the fit curves or use different colors?

As Reviewer 2 pointed out, there is another observation of Venus, namely the attempt to detect phosphine in the Venus atmosphere. These measurements were done in parallel (with the other frequency channel of upGREAT) to our measurements. Although phosphine was not detected it is also an observation of Venus. We

explained that in lines 323 - 325. Since it does not add new information about the Venus atmosphere, we prefer not to mention it in the main text. We have changed Fig.M1 (different colors for fit and measurement).

I.385: “the atmospheric temperature does not change much”. Please, add a reference.

We added a reference (Limaye et al., Ref. 12).

Overall, this is a very interesting work, with promising results and I wonder whether if you have considered studying the correlation between your atomic oxygen column density and the O₂ nightglow observed by Venus Express? Or, to compare your atomic oxygen temperature and the O₂ nightglow? These are two observational datasets that do not require any modeling and it would definitely be a good way to connect them and bring this work a step further.

We have considered the correlation between atomic oxygen column density and O₂ nightglow observations by Venus Express. In lines 53-63 we briefly summarize the correlation between both and how O₂ nightglow data from Venus Express are used to derive atomic oxygen column densities (references are given, Refs: 1, 2, 3, 8). We would like to point out that in these references the nightglow data are used to derive the atomic oxygen density and the focus is on atomic oxygen. For example, Ref. 2 provide no map of the nightglow but an atomic oxygen map. In addition, for a detailed comparison one would need nightglow data from the same date (or close to that date) as our observation. These data are not available, because Venus Express stopped operation in 2014. The importance of using data from the same date is shown in Ref. 1 by Gérard et al. where the authors compare O₂ nightglow data with atomic oxygen data from the same orbit of Venus Express (at 00:30 LT). Conclusions such as for example in Gilli et al. (Ref. 8) that atomic oxygen data are in good agreement with O₂ nightglow data are, therefore, not possible with our observations simply because of the lack of nightglow data measured at the same time. Despite that, we compare our data with atomic oxygen data obtained from nightglow observations with Venus Express (lines 172 – 192). Here, we explicitly refer to O₂ nightglow measurements of Gérard et al (Ref. 29). We have only nighttime data until 21 LT. Therefore, the comparison is limited, since nightglow data are most pronounced around the antisolar point.

Minor comments

I.23: I suggest you add “However, past and current detection methods”. **Done**

I.27: “on board SOFIA”. **Done**

L.26-27: Maybe the upGREAT and SOFIA acronym should be explained in the abstract. **Done**

I.28: please, add “0.7 to 3.8 x 10¹⁷ cm⁻² between 15:00 and 21:00 Local Time”. **Done**

I.32: “These data are the first dayside measurements of atomic oxygen.” It should appear sooner in the abstract. Maybe, I.25 “We report on the first direct detection...”.

We deleted the sentence “These data are the first ...” and added in line 25 “first”.

I.32: “This new method” **Done**

I.39-40: please, add commas “On the nightside, atomic oxygen recombines in a three-body reaction to molecular oxygen. This is the main source of excited oxygen molecules, which, in turn, are the source of the Venus nightglow.” **Done**

I.43: what does “This” refer to? The global circulation?

We rephrased the sentence. It now reads: “In addition, atomic oxygen can be used as tracer for the global circulation in the upper thermosphere (~130 – 250 km) as demonstrated by measurements of the oxygen dayglow with the Pioneer Venus Orbiter.” (lines 45 -47)

I.48: “limb” Do you mean terminator?

We mean the edge of the disc of Venus. We changed “limb” to “edge”.

I.51: “onboard” instead of “on” Done

L.52: please, provide the transition of the 1.27 μm airglow. Done

I.54: add comma “quenching coefficients, reaction rates and efficiencies, atomic oxygen densities have been” Done

L.55: add reference 3 together with reference 2 Done

I.57: change “longitude” to “latitude” Done

I.59: “stronger”: please, quantify. Done

I.68: add a comma after “In the atmosphere of Venus” Done

I.74-75: the acronym should probably be explained in the abstract. Done

I.75: “on the SOFIA airplane”. And, also, add a reference to Methods/SOFIA upGREAT observations. Done

I.80: add a comma after “In total”. Done

I.79-80: the sentence is a bit long. I suggest “high spectral resolving power of upGREAT allows distinguishing both. The telluric atomic oxygen line is used for frequency and radiometric calibration.”

We have implemented this suggestion. (lines 82 - 84)

I.81: replace the comma after “measured” by “:”. Done

I.107: please, change to 246 K, for consistency with the abstract and the rest of the manuscript. Done

I.114-115: replace “which are on” by “in”. Done

I.115: a space should be added between “points” and “(circles)”. Done

I.123-124: avoid the repetition of the word “measurements” in the same sentence.

We replaced the second “measurements” by “observations”.

I.125: “predicts” sounds like the result of a model, while Sagawa (2008) corresponds to an observational study. Maybe “observes” or “deduces” are more appropriate terms here.

As suggested, we introduced “deduces”.

I.136: “This is determined from **the fit of** a radiative transfer model”. Done

l.138: change “ref 16” to the names of the authors, as done in the rest of the manuscript. **Done**

L.158: the reference “2” to Brecht et al is a little bit weird at its current location. It looks like a squared value. Figure 3: For clarity, it would be nice to add more clearly some local time values (12:00, 18:00 and 24:00 LT, at least).

We have moved reference “2” to a different position in the sentence. We have added local times and latitudes to Fig. 3.

l.176: use a period instead of a colon. **Done**

l.179: replace “towards the nightside” by “near the terminator”. **Done**

l.130: “Future observations, **especially near the antisolar and subsolar points but also at all solar zenith angles**, will provide...” **Done**

l.227: ref 17 is not cited in the text.

It was cited in line 107 of the original manuscript. In the revised version it is cited in line 112 (Ref. 22).

l.282: please, add commas “The nominal pointing of the SOFIA telescope, as defined by star tracker cameras, was ...” **Done**

Please, add commas after “half on Venus” (l. 291), “For further analysis” (L.305), “Note that” (l.306), “for Venus” (l. 306), “Throughout the analysis” (l. 313), “Here” (L. 325), “the sky” (l.343), “In both cases” (l.344), “center of the line” (l.354), “above” (L.358), “For each layer” (l.361), “input” (l.386). **Done**

L.312: please, add a reference to the Earth’s study. **Ref. 11 (Richter et al.) is added.**

L.317: I think it should be equation (1) instead of (2). Same for the following equations.

Yes, the numbering was wrong. We corrected it.

l.329: $S=1.131 \times 10^{-21}$ (x is missing) **“x” is added.**

l.346: the sentence is a bit long. I suggest “... Earth atmosphere at the altitude where the line becomes optically thick. In this case, it corresponds to the lower thermosphere, at around 100 km altitude.”

We changed the sentence as suggested.

Reviewer #2 (Remarks to the Author):

Overall:

The manuscript is discussing a unique dataset that was obtained by the upGREAT heterodyne spectrometer on board SOFIA. Atomic oxygen is an important chemical species in atmospheres but can also be difficult to directly observe. This manuscript is providing a decent and brief overview of the importance of atomic oxygen in the Venusian atmosphere. It also does a good job in comparing their results to past observations and model simulations. There are a few revisions being suggested that are easily corrected. The recommendation is this manuscript needs minor revisions before publication.

Major comments:

(1) The introduction/motivation section (~Lines 38 – 44) needs to be re-organized and focused. Things are briefly touched on but the connection between the points are not clear or fully covered with appropriate references. It is unclear if the authors want to only discuss one nightglow (O2 IR) or briefly state a few.

Atomic oxygen is important for photochemistry, circulation, and energy budget. Due to a lack of direct atomic oxygen observations, scientists have had to rely on other atmospheric features to understand the atomic oxygen distribution, such as nightglow. Nightglows (e.g. O2 IR, O2 visible, NO UV) provide insight into the photochemistry and dynamics due to the intensity and location of the features on the nightside

[e.g. Alexander et al. 1993 (<https://doi.org/10.1029/93JE00538>);
Bougher et al. 1990 (<https://doi.org/10.1029/JA095iA05p06271>);
Bougher et al. 1994 (<https://doi.org/10.1029/93JE03431>);
Garcia Munoz et al. 2009 (<https://doi.org/10.1029/2009JE003447>);
Brecht et al. 2011 (<https://doi.org/10.1029/2010JE003770>);
Gerard et al. 2017 (<https://doi.org/10.1007/s11214-017-0422-0>);
Navarro et al. 2021 (<https://doi.org/10.1016/j.icarus.2021.114400>)].

Understanding atomic O distribution is greatly important for CO2 15 micron cooling (the dominate cooling mechanism in the upper atmosphere), thus important for energy balance. [e.g. Bougher et al. 1994 (<https://doi.org/10.1029/94JE01088>); Gilli et al. 2021 (<https://doi.org/10.1016/j.icarus.2021.114432>)]

We have reorganized the introduction. Since we do not want to discuss the airglows, we have omitted the sentence “This is the main source of excited oxygen molecules, which, in turn, are the source of several Venus nightglows”. Now, in lines 38 to 41 the generation and destruction of atomic oxygen is described. In lines 41 to 45 the importance of atomic oxygen for the photochemistry and energy balance are briefly explained. Finally, in lines 45 to 47 the role of atomic oxygen as tracer for motion in the atmosphere is mentioned. We have added several of the references, which are suggested by the reviewer (Bougher et al. 1994, Gilli et al. 2021, Gerard et al. 2017, Alexander et al. 1993).

(2) Throughout the manuscript, please be clearer on the local time being discussed. There are places where it isn't clear if the full range of local times are being discussed, the dayside, or the nightside are being discussed.

We have added this information (see lines 136, 150, 154, 165).

(3) The radiative transfer (RT) code needs more support. It is unclear if the RT code has been published before and how the input values were determined. This section needs references and/or more detail.

We have added a section with references (lines 355 to 361) on the RT code, explaining, in particular, how the verification of the code was done. The input parameters for analyzing the telluric line are given in lines 407 – 418. The input parameters for analyzing the Venus line are given in lines 430 -440. With this input parameters and with equations (1) – (3) as well as the accompanying explanation the radiative transfer code is completely described.

Details:

Line 20: replace “is mostly located” with “peaks” Done

Line 41: replace “source of the Venus nightglow” with “source of several Venus nightglows”. O2 IR nightglow and the O2 Visible nightglow can be produced from the 3-body reaction.

We replaced it as suggested by the reviewer.

Line 39 – 44: This group of sentences needs rephrasing and re-organization. See major comment #1 Done, see answer to major comment no. 1.

Line 44: “Pioneer” -> “Pioneer Venus Orbiter” Done

Line 50: State the wavelength or just “IR” after “O2” We added the wavelength.

Line 53-55: This sentence is missing a word or punctuation to provide more clarity in the statement “reaction rates and efficiencies atomic oxygen densities have been calculated.”

A comma is added after “efficiencies” (as suggested by Reviewer 1).

Line 54: Is temperature a-priori information?

No, it is obtained from measurements. We have added “temperature” in line 56.

Line 79: it is unclear what is meant by “allows distinguishing both and using the telluric...”

We have rephrased the sentence as suggested by Reviewer 1, It now reads: “... high spectral resolving power of upGREAT allows distinguishing both. The telluric atomic oxygen line is used for frequency and radiometric calibration.”

Line 138: check on how to write reference within statement We changed it to “Limaye et al.”

Line 140: “The average nightside temperature is lower”... what is it lower than?

It is lower than the dayside temperature. Now the sentence reads “is lower than the dayside temperature.”

Line 145: remove parentheses from around the temperature values. Done

Line 151: Please clarify the column density value stated on line 152 is for the dayside.

We rephrased the sentence and added “dayside”. Now it reads: “The column density derived from this dayside spectrum at 15:36 LT is ...” (line 165 - 166).

Line 154: Please clarify “These values”. Is this regarding the range or the average or all of the above?

We replaced “These values” by “Our values” (line 172).

Line 155: “O2 nightglow”, do you mean “O2 IR nightglow”? We added “infrared”.

Line 156: Please clarify “these data”. It is unclear if this refers to the new observations or from previous work. Also unclear which local time this refers to.

We have reworded the sentence. Now it reads “obtained from these nightglow data” (line 174).

Line 157: A clearer statement might be: “The column densities derived from VIRTIS altitude profiles as reported in Brecht et al. 2012 are lower ($\sim 1 \times 10^{17} \text{ cm}^{-2}$).”

We have changed the sentence as suggested by the reviewer.

Line 157-158: Brecht et al. 2012 report GCM simulation results too. Should state those values to compare with the VIRTIS results, your results, and with the next sentence about a different GCM simulation result.

We now refer to the global circulation model of Brecht et al. (line 177) and we have extended the discussion in this paragraph.

Line 158: missing period at end of sentence Done

Line 159: rephrase; “Also, results from a different model yielded column densities around $2 \times 10^{17} \text{ cm}^{-2}$ which are comparable with our observations.”

We changed the sentence according to the reviewer’s recommendation.

Line 168 – 170: (Fig. 3 Caption) Please state if this is the evening or morning terminator

It is the evening terminator. We added it to the figure caption.

Line 283-284: Please clarify this statement. There are other SOFIA observations of Venus. Most of them are conference proceedings, but Cordiner et al. 2022 (<https://doi.org/10.1029/2022GL101055>) is actually published.

Yes indeed, the measurements reported by Cordiner et al. were made during exactly the same SOFIA flights as our measurements. Cordiner et al. did not observe the phosphine emission, which was the aim of their measurements but derived upper limits of the phosphine concentration in the Venusian atmosphere. We have corrected our statement accordingly. Now it reads: “It is worth mentioning that these measurements are the first observation of a gaseous line in the Venusian atmosphere with SOFIA. Our observation took place during the same SOFIA flights as the unsuccessful attempt to detect phosphine in the Venusian atmosphere⁷.” We have referenced the work of Cordiner et al (Ref. 7 of the methods section).

Line 291: Please clarify or rephrase; “In order to avoid that most of the pixels...” What is being avoided?

We have reworded the sentence in order to be more clear (lines 331 - 333). Now it reads: “In order to avoid having most of the pixels half on and half off Venus, the central pixel of the mixer array was pointed on the dayside at about 15:00 LT. In this case, ...”

Line 305: a comma is needed after, “For further analysis” Done

Line 311: Is there a reference for the radiative transfer code? If the code is not public and/or published, please add supporting references for values etc.

We have added more information about the radiative transfer code (lines 354 – 361) and its verification and we added a reference (Ref. 11, Richter et al. 2021).

Line 358: Please rephrase this sentence. What temperature is determined?

It is the temperature of the Earth atmosphere where the atomic oxygen line becomes optically thick. Now it reads “the temperature of the Earth atmosphere at which the line saturates”.

Line 396 – 399: Could you please provide more information, such as key words to put in the search fields to find the exact data that was used? As written, it is unclear if the data is available.

We have added the following key words: Search terms are: “Spatial Constraints: Solar System Target”, Object name: Venus”, “Proposal Constraints: Primary Investigator: Heinz-Wilhelm Huebers, Plan ID: 75_0068, Observation Date: 2021-11-10 to 2021-11-13”, “Instrument Constraints: GREAT”, “Data Product Constraints: Level 1”.

REVIEWER COMMENTS

Reviewer #1 (Remarks to the Author):

Thank you for improving the manuscript according to the reviewer's comments.

The paper is now almost ready for publication.

However, I have an additional comment to make to figure 2b. This figure very well illustrates the variations in the dayside according to SZA. The nightside observations, on the contrary, have been averaged into one single point. According to Tables 2 to 4, seven observations have been acquired between 50 and 84° in the dayside and shown in Figure 2b as individual data points. Ten observations have been made in the nightside for SZA ranging from 90 to 120°. So, why do you average them? It would be much better to see all these points as well. I suggest you add all the nightside data points according to the SZA, to see if an increase towards the antisolar point is also observed. It will thus be much easier to compare these results with those from the Venus Express mission.

Regarding the explanation the authors added about the variability observed near the terminator (l. 183-192), they mention the work of Soret et al. (2012) and the substructure in the nightglow near the antisolar point. I suggest you also refer to the Soret et al. (2014)'s paper that showed that the nightglow is highly variable and that local maxima can also occur in regions located near the terminator (see figure 4 for example), which is a confirmation of the author's statement.

The reference can be found here: <http://dx.doi.org/10.1016/j.icarus.2014.03.034>

Reviewer #2 (Remarks to the Author):

The authors have improved the manuscript and addressed my concerns. The recommendation is this manuscript needs one revision before publication.

Line 83: not a complete sentence, maybe “allows for the ability to distinguish both.”

Berlin, September 4, 2023

Dear Reviewers:

Once again thank you for your thoughtful comments. We have taken them into account in this revision. In the red-line version of our manuscript (Huebers-Venus-Revision2-red-line-2023-09-04.pdf) these changes are indicated in red. Below you will find comments on your remarks (in red). We hope that the revised manuscript meets your expectations.

Sincerely,

Heinz-Wilhelm Hübers (on behalf of the authors)

REVIEWER COMMENTS

Reviewer #1 (Remarks to the Author):

Thank you for improving the manuscript according to the reviewer's comments.

The paper is now almost ready for publication.

However, I have an additional comment to make to figure 2b. This figure very well illustrates the variations in the dayside according to SZA. The nightside observations, on the contrary, have been averaged into one single point. According to Tables 2 to 4, seven observations have been acquired between 50 and 84° in the dayside and shown in Figure 2b as individual data points. Ten observations have been made in the nightside for SZA ranging from 90 to 120°. So, why do you average them? It would be much better to see all these points as well. I suggest you add all the nightside data points according to the SZA, to see if an increase towards the antisolar point is also observed. It will thus be much easier to compare these results with those from the Venus Express mission.

We have plotted the individual data points and deleted the statement about the average value in the main text (new Fig. 2b). There is no increase towards the antisolar point, probably because the measurements are too far away from the antisolar point and dynamical processes, which may lead to a local maximum of atomic oxygen (as discussed in lines 190 – 195). We have added the following sentences: "Note that sunset at the equator occurs at 100 km altitude at 101.3° and 18:41 LT." (end of Fig. caption 2) and "At nighttime no trend of the column density is observable" (line 173).

Regarding the explanation the authors added about the variability observed near the terminator (l. 183-192), they mention the work of Soret et al. (2012) and the substructure in the nightglow near the antisolar point. I suggest you also refer to the Soret et al. (2014)'s paper that showed that the nightglow is highly variable and that local maxima can also occur in regions located near the terminator (see figure 4 for example), which is a confirmation of the author's statement. The reference can be found here:

<http://dx.doi.org/10.1016/j.icarus.2014.03.034>

Thank you for this suggestion. We have added a sentence about the variability of the oxygen nightglow and we

have added the reference (reference no. 30). The new sentence is: "It is worth noting that Soret et al. have identified a substructure of the antisolar atomic oxygen maximum from observations of the 1.27- μm nightglow of O_2 as well as highly variable, local O_2 maxima in regions close to the terminator^{3,30}." (lines 193 - 195).

Reviewer #2 (Remarks to the Author):

The authors have improved the manuscript and addressed my concerns. The recommendation is this manuscript needs one revision before publication.

Line 83: not a complete sentence, maybe "allows for the ability to distinguish both."

Reviewer #1 (Remarks to the Author):

We have corrected this mistake. Now the sentence reads "The high spectral resolving power of upGREAT allows for the ability to distinguish both." (lines 82- 83).

REVIEWERS' COMMENTS

Reviewer #1 (Remarks to the Author):

Thank you for taking the reviewers remarks into account.

The manuscript is now ready for publication.